# Magnetoresistance of Ultralow-Hole-Density Monolayer Epitaxial Graphene Grown on SiC

**DOI:** 10.3390/ma12172696

**Published:** 2019-08-23

**Authors:** Chiashain Chuang, Chieh-Wen Liu, Yanfei Yang, Wei-Ren Syong, Chi-Te Liang, Randolph E. Elmquist

**Affiliations:** 1Department of Electronic Engineering, Chung Yuan Christian University, Taoyuan 32023, Taiwan; 2National Institute of Standard and Technology (NIST), Gaithersburg, MD 20899, USA; 3Department of Physics, Case Western Reserve University, Cleveland, OH 44106, USA; 4Department of Physics, National Taiwan University, Taipei 106, Taiwan

**Keywords:** silicon carbide, epitaxial graphene, magnetoresistance, PMOS, quantum hall, resistance standard

## Abstract

Silicon carbide (SiC) has already found useful applications in high-power electronic devices and light-emitting diodes (LEDs). Interestingly, SiC is a suitable substrate for growing monolayer epitaxial graphene and GaN-based devices. Therefore, it provides the opportunity for integration of high-power devices, LEDs, atomically thin electronics, and high-frequency devices, all of which can be prepared on the same SiC substrate. In this paper, we concentrate on detailed measurements on ultralow-density *p*-type monolayer epitaxial graphene, which has yet to be extensively studied. The measured resistivity *ρ*_xx_ shows insulating behavior in the sense that *ρ*_xx_ decreases with increasing temperature *T* over a wide range of *T* (1.5 K ≤ *T* ≤ 300 K). The crossover from negative magnetoresistivity (MR) to positive magnetoresistivity at *T* = 40 K in the low-field regime is ascribed to a transition from low-*T* quantum transport to high-*T* classical transport. For *T* ≥ 120 K, the measured positive MR ratio [*ρ*_xx_(*B*) − *ρ*_xx_(*B* = 0)]/*ρ*_xx_(*B* = 0) at *B* = 2 T decreases with increasing *T*, but the positive MR persists up to room temperature. Our experimental results suggest that the large MR ratio (~100% at *B* = 9 T) is an intrinsic property of ultralow-charge-density graphene, regardless of the carrier type. This effect may find applications in magnetic sensors and magnetoresistance devices.

## 1. Introduction

Silicon carbide (SiC) is an extremely useful semiconductor material which has already found applications in light-emitting diodes (LEDs) [1], detectors [2], and power devices [3] that operate at high temperatures and/or high voltages. SiC is also an excellent substrate for growing GaN-based materials [4] and, most importantly, for preparing wafer-scale epitaxial graphene using the high-temperature sublimation technique [5]. Therefore, it is possible to prepare not only low-dimensional graphene-based electronic devices [5] but also high-power devices [3], LEDs [1], and GaN-based high-electron mobility transistors (HEMTs) [6] on the same SiC substrate.

Graphene, which is a single layer of carbon atoms bonded in a honeycomb lattice, has continued to attract great interest because of its fundamental importance as well as its practical device applications [7,8,9]. Although graphene prepared by mechanical exfoliation is of high-quality [7,8,10], the limited size of the resulting flakes may be hindered in real-world applications. Graphene prepared by chemical vapor deposition (CVD) [11,12,13] can be of meter-size, yet the subsequent transfer required for device fabrication may introduce undesired polymer residues and wrinkles, which may compromise its device performance. In contrast, monolayer epitaxial graphene (EG) grown on SiC [5,14,15,16,17] can be of wafer-size, does not require any transfer process, and has already found applications in high-frequency transistors [18] and for maintaining the quantum Hall resistance standards [15,19,20]. The epitaxial carbon interface layer that forms on SiC (0001) is a precursor to true conducting graphene, and interfacial covalent bonding conveys strong *n*-type doing to the first conducting layer of EG. To date, *n*-type EG grown on SiC has been extensively studied in the context of magnetic sensors [21], the insulator-quantum Hall transition [22,23,24,25], interaction effects enhanced by disorders [26,27], and so on. However, there is a dearth of study on *p*-type graphene on SiC. It is the purpose of this work to present detailed transport measurements on ultralow-density *p*-type epitaxial graphene grown on SiC. We will present resistivity measurements over a wide range of temperature 1.5 K ≤ *T* ≤ 300 K and discuss the pronounced positive magnetoresistivity effect, which persists up to room temperature and may find applications in magnetic sensing and magnetoresistance devices. Our new experimental results on ultralow-hole-density monolayer graphene may well lay the foundation for future graphene-based complimentary metal-oxide-semiconductor (CMOS) technology, in which both *n*-type MOS (NMOS) and *p*-type MOS (PMOS) are fabricated on the same SiC substrate.

## 2. Materials and Methods

Our epitaxial monolayer graphene was grown on a chemically and mechanically polished 6H-SiC (0001) substrate at 1850 °C for 30 min under an Ar gas pressure at 98 kPa [17,20]. Figure 1a,b show atomic force microscopy (AFM) results of the height and phase measurements. We could see narrow terraces and bilayer graphene near the step edges formed on the SiC substrate. Such terrace growing direction was almost perpendicular to the source and drain current direction since our AFM tip scanning direction was parallel to the source and drain current direction, which was a better contact design due to the anisotropic quantum Hall effect, in order to observe flat Hall plateau and positive magnetoresistance [28]. Interestingly, such monolayer-bilayer junctions or carbon buffer layer in EG typically reveal pronounced *p*-type or *n*-type doping, ranging from 10^12^ to 10^13^ cm^−2^, which is highly consistent with our experimental results [29,30,31]. In this work, we did not get to measure the anisotropic transport perpendicular/parallel to the terrace in our EG devices, which is an interesting subject warranting further investigation in the future. In order to prevent contamination from photoresist, we deposited a bilayer of 5 nm thick Pd and 10 nm thick Au on the as-grown EG [17]. We performed standard optical lithography to fabricate Hall bar devices with source and drain length *L* = 600 μm and width *W* = 100 μm [17]. The protective metal was removed from the Hall bars using dilute aqua regia in the final step, a process that can produce low-electron-density epitaxial graphene devices [17]. Since air and moisture usually introduce *p*-doping, ultralow-hole-density graphene can be obtained if the low-electron-density device was left in an ambient condition over a long period of time (e.g., ~3 months). However, the carrier densities of our low-carrier-density EG devices were not stable in air. Therefore, we needed to measure the *p*-type devices in a cryostat, which only exposed to helium gas/liquid without air molecule absorption so as to keep the *p*-type device stable for a long period of time (over two months). However, the carrier density in our low-carrier-density EG devices were easily variable in the air condition. Therefore, we needed to measure all the data at once in the *p*-type or *n*-type devices in a vacuum cryostat without air molecular absorption in a very long time period. Four-terminal longitudinal resistivity, *ρ*_xx_ and Hall resistance *ρ*_xy_ were measured using standard ac lock-in techniques.

## 3. Experimental Results

Figure 2 shows $\rho_{xx}$ and *ρ*_xy_ as a function of magnetic field at *T* = 1.5 K. From the low-magnetic field Hall slope, we estimate the hole density to be *p* = 6.7 × 10^9^ cm^−2^ and the Hall mobility to be *μ*_H_ = 2600 cm^2^/Vs. In the high-magnetic-field regime, $\rho_{xx}$ remains high so that dissipationless transport is not observed. *ρ*_xy_ does not approach the quantized value of *h*/(2*e*^2^) and tends to decrease with the increasing magnetic field. Such peculiar behaviors could be possibly attributed to the electron-hole puddles due to very low carrier density near Dirac’s point because of the intrinsic bilayer inclusions near EG terraces so as to cause strong carrier scattering and not follow dissipationless transport behaviors [32,33]. It is possible that the contacts introduce *n*-doping underneath so that the *p*-doped device does not show good quantum Hall steps (pn junction effect).

In order to further study our *p*-type epitaxial monolayer graphene, we carried out extensive low-field magneto-transport measurements over a wide range of temperature. As shown in Figure 3a, activated behavior [34,35,36,37] was not applicable. The linear fits over a small range of temperature suggest that Mott variable range hopping (VRH) [37] and Efros–Shklovskii VRH [38] may be applicable to our experimental results. Some evidence for the localization transport mechanism [39], such as $\rho_{xx}$, shows a ln*T* dependence in the low-temperature regime, as presented in Figure 3e. At present, we cannot underpin the dominant transport mechanism (VRH or localization) in our *p*-doped EG device. In order to further study this insulating behavior, we plotted $\ln\rho_{xx}$ as a function of 1/*T*, *T*^−1/3^ and *T*^−1/2^ in Figure 3b–d, respectively. In all cases, one could only obtain a linear fit over a small range of temperature, suggesting that activated behavior [36], Mott variable range hopping (VRH) [37], and Efros–Shklovskii VRH [38] may be applicable to our experimental results in a high temperature regime. However, the transport mechanism was localization [39], as $\rho_{xx}$ shows a ln*T* dependence in the low-temperature regime, as presented in Figure 3e.

As shown in Figure 2 and Figure 4a, negative magnetoresistivity (NMR), in the sense that *ρ*_xx_ decreases with the increasing magnetic field, is observed in an ultra-low-hole carrier density regime. Such NMR is commonly observed in monolayer epitaxial graphene and is sometimes ascribed to suppression of weak localization (WL). However, in our case, $\rho_{xx}$ dropped rapidly below 4 kΩ/□ at 1.5 K as shown in Figure 4a so that the conventional WL model [40] was not valid. The observed NMR, which shows a strong temperature dependence, is tentatively ascribed to quantum transport [41]. Such quantum transport and NMR behaviors could be described by Mott VRH, Efros–Shklovskii VRH, and localization that were observed in our results.

As shown in Figure 4a, at *T* = 40 K in the low-field regime (−0.3 T ≤ *B* ≤ 0.3 T), there is a crossover from NMR to positive magnetoresistivity (PMR) [41], which was also observed in an ultra-low-electron carrier density regime [21]. By PMR we mean that the resistivity increases with the increasing magnetic field. Interestingly, the observed PMR becomes strongest at around *T* = 110 K and persists up to *T* = 300 K (Figure 4b), indicating that its physical origin is classical transport. In our case, the PMR is believed to be due to density inhomogeneity in our ultralow-density monolayer graphene, since the magnetoresistivity ratio decreases with increasing temperature, consistent with the classical Parish and Littlewood model [26,42,43]. The observed NMR-PMR crossover can be attributed to a transition from quantum transport to classical transport with increasing temperature [41].

In the field of magnetic sensing devices, the magnetoresistivity (MR) ratio is defined as [$\rho_{xx}$(*B*) − $\rho_{xx}$(*B* = 0)]/$\rho_{xx}$(*B* = 0). Figure 5 shows the MR ratio at various temperatures. We can see that for *T* ≥ 120 K, the MR ratio decreases with the increasing temperature, suggesting that our ultralow-hole density graphene serves as a better magnetic sensing device at a low temperature (MR ratio around 30% at a *T* = 120 K). The respectable MR ratio (about 20%) at 2 T at room temperature suggests that our ultralow-hole density graphene may find applications in magnetic sensing. In the low-temperature regime (*T* = 40 K and *T* = 50 K), there is a crossover from PMR to NMR; therefore, this device is not suitable for MR applications.

Previously, we have shown that, for example, (in our experiment), using the low-pressure gentle heating technique, we are able to remove air adsorbate on epitaxial graphene so that chemical doping can be tuned [44]. By heating the graphene device at 323 K for 5 min in a subsequent cool-down to 1.5 K, a crossover from ultralow *p*-type to ultralow *n*-type (*n* = 8.7 × 10^9^ cm^−2^ and *μ*_H_ = 10,400 cm^2^/Vs) occurs, as shown by the *ρ*_xy_ data in Figure 6. Compared to the data shown in Figure 2, $\rho_{xx}$ is now much lower at high fields and approaches 0, and *ρ*_xy_ approaches a quantized value of *h*/(2*e*^2^) in the high-field regime. Since the data shown in Figure 2 and Figure 6 were taken on the same device, the MR characteristics observed on the electron side unequivocally indicate monolayer epitaxial graphene is present (our low-*T* heating technique only removes gas adsorbate and thus does not vary the graphene structure).

Recently, we have shown that ultralow-electron-density epitaxial monolayer graphene can be used as a magnetic sensing device and its MR ratio is around 100% at 9 T at room temperature [31]. It is therefore natural for us to see whether our ultralow-hole-density monolayer graphene can also serve as a magnetic sensor. Despite the vast differences between the *n*- and *p*-type graphene in the low-temperature and high-magnetic field regime, as shown in Figure 2 and Figure 6, interestingly, almost identical MR ratio and behavior on both the electron and hole sides are observed (Figure 7). Our results suggest that the observed large MR ratio at room temperature is an intrinsic property of ultralow-density graphene, regardless of its charge type.

## 4. Conclusions

In conclusion, we have reported extensive magneto-transport measurements on ultralow-hole-density epitaxial monolayer graphene grown on SiC. Interestingly, over the whole measurement range, the resistivity of the *p*-type monolayer graphene shows insulating behavior. The most likely transport mechanism is localization in the low-temperature regime. In the low magnetic field regime (−0.3 T ≤ *B* ≤ 0.3 T), there is a crossover from negative magnetoresistivity to positive magnetoresistivity at *T* = 40 K, which is ascribed to a transition from quantum to classical transport with increasing temperature. The positive magnetoresistivity is most pronounced at around *T* = 120 K and becomes weaker at higher temperatures; however, PMR persists up to 300 K, indicating that ultralow-hole-density monolayer graphene may find applications in magnetic sensing and magnetoresistance devices. Simply by gentle heating in a vacuum, we can tune our ultralow-*p*-type device to an ultralow-*n*-type one. Interestingly, at room temperature, almost an identical magnetoresistivity ratio is observed on both the hole and electron sides up to 9 T, indicating that the observed large MR is an intrinsic property of an ultralow-density monolayer graphene, regardless of whether it is *n*-type or *p*-type. Our experimental results may open the door for future graphene-based CMOS technology on SiC with hexagonal boron nitride as a top dielectric spacer [43].

## Figures and Tables

**Figure 1 materials-12-02696-f001:**
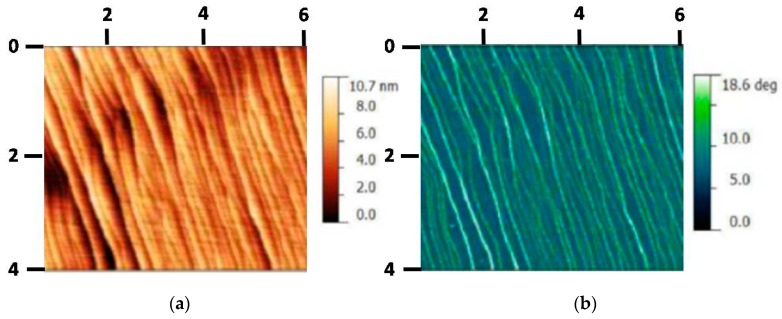
Atomic force microscopy (**a**) height and (**b**) phase measurements.

**Figure 2 materials-12-02696-f002:**
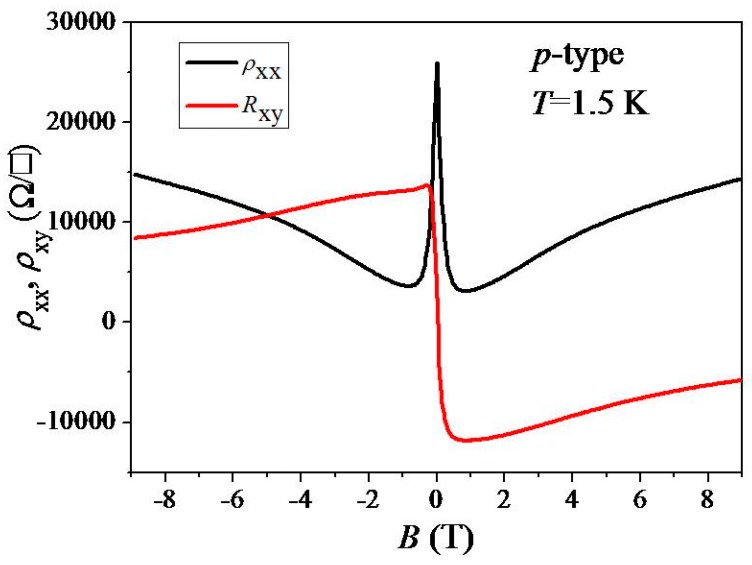
$\rho_{xx}$ and *ρ*_xy_ as a function of magnetic field *B* at *T* = 1.5 K (*p*-type).

**Figure 3 materials-12-02696-f003:**
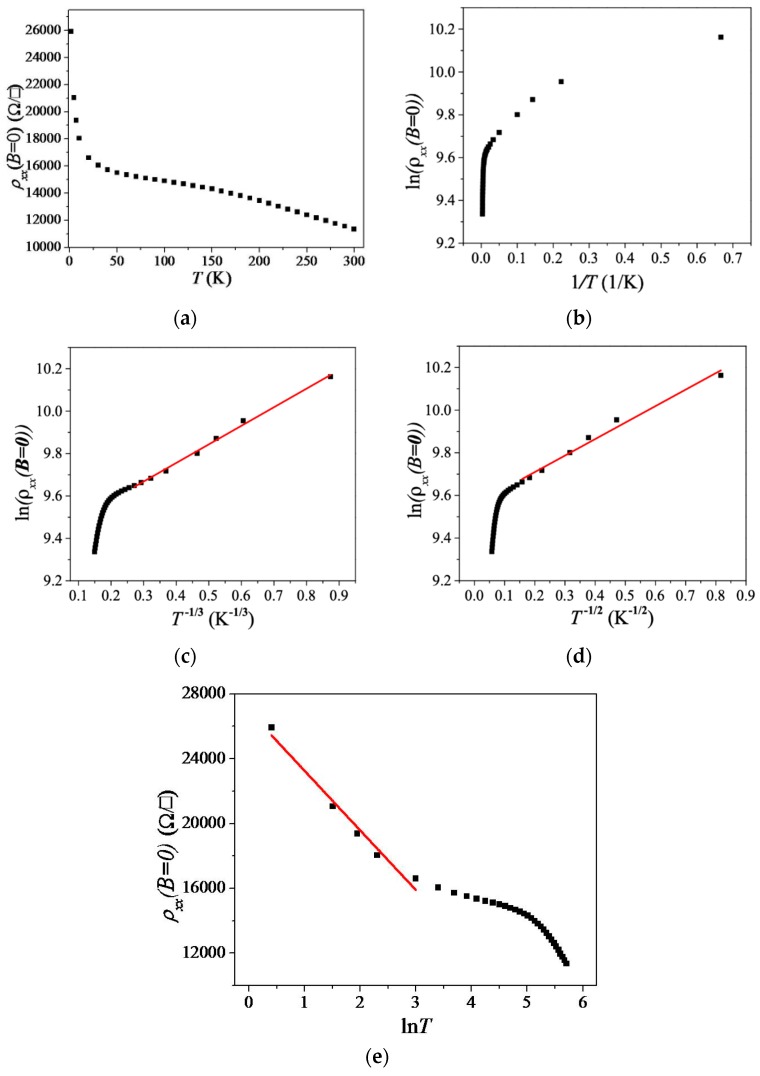
(**a**) $\rho_{xx}$ as a function of *T*. (**b**) $\ln\rho_{xx}$ as a function of 1/*T*. (**c**) $\ln\rho_{xx}$ as a function of *T*
^−1/3^. The red line shows a fit to the data over a small range of $\ln\rho_{xx}$. (**d**) $\ln\rho_{xx}$ as a function of *T*
^−1/2^. The red line shows a fit to the data over a small range of $\ln\rho_{xx}$. (**e**) $\rho_{xx}$ as a function of ln*T*. The red line corresponds to a linear fit to the experimental data in the low-*T* regime.

**Figure 4 materials-12-02696-f004:**
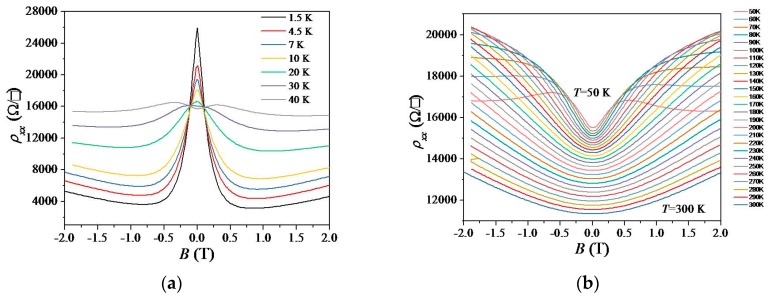
(**a**) $\rho_{xx}$ (*B*) at various temperatures *T* (low-*T* regime) in an ultra-low-hole carrier density regime. (**b**) $\rho_{xx}$ (*B*) at various temperatures *T* (high-*T* regime).

**Figure 5 materials-12-02696-f005:**
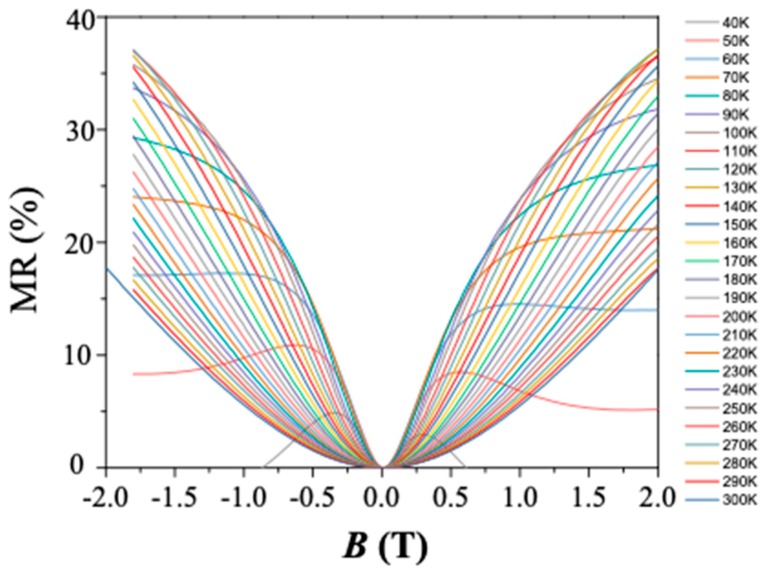
Magnetoresistivity (MR) ratio at various temperatures *T*.

**Figure 6 materials-12-02696-f006:**
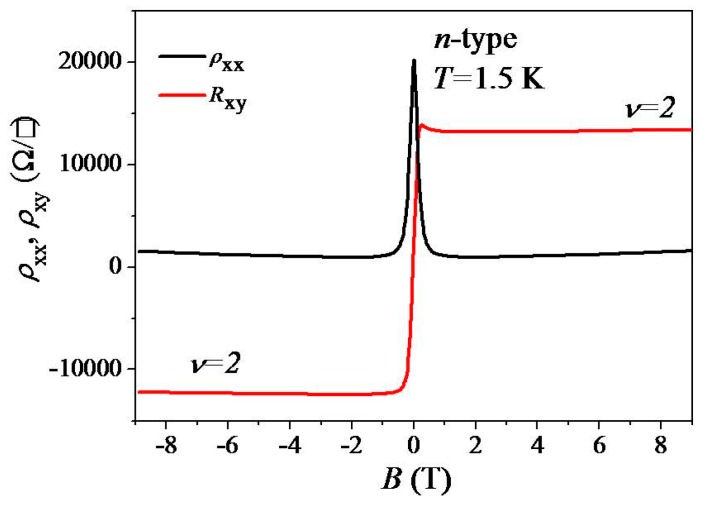
$\rho_{xx}$ and *ρ*_xy_ as a function of magnetic field *B* at *T* = 1.5 K (*n*-type; after low-*T* gentle heating).

**Figure 7 materials-12-02696-f007:**
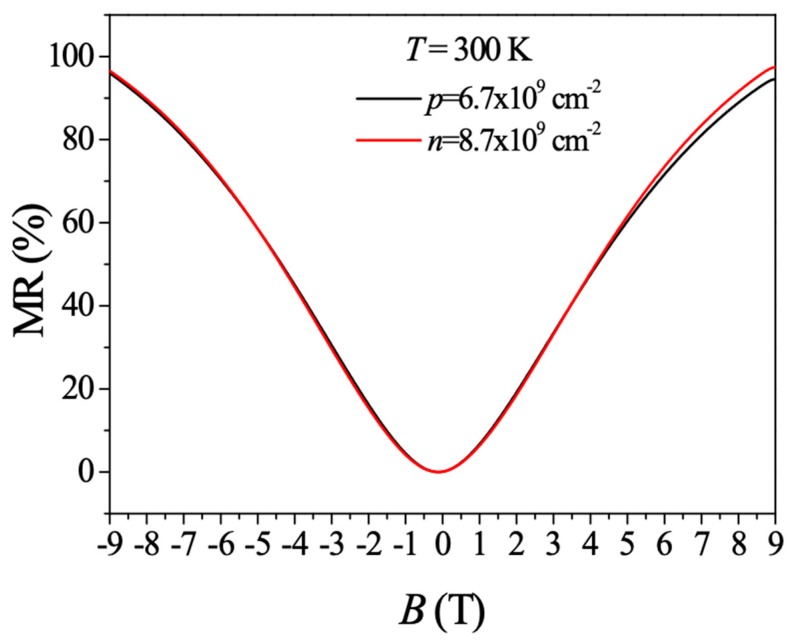
MR ratio for both ultralow-*n-* and ultralow-*p*-density monolayer graphene at room temperature.

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
