# Peer review of "Magnetoresistance of Ultralow-Hole-Density Monolayer Epitaxial Graphene Grown on SiC"

_materials, 2019, doi:10.3390/ma12172696_

Round 1
Reviewer 1 Report
This papers reports detailed magneto-transport measurements on monolayer epitaxial graphene with ultra-low hole density grown on silicon carbide. For measurements performed at low magnetic fields (below 0.3 T), a crossover from negative magnetoresistivity to positive magnetoresistivity was observed at a temperature T=40 K, which was ascribed to a transition from quantum to classical transport. The positive magnetoresistivity persisted up to 300 K, indicating possible applications of the ultralow-hole-density monolayer graphene in magnetic sensing and magnetoresistance devices. The ultralow-p-type graphene was converted into ultralow-n-type one by gentle heating in vacuum. Almost identical magnetoresistivity ratio (MR) was found in both cases, indicating that the observed large MR is an intrinsic property of an ultralow-density monolayer graphene, regardless whether it is n-type or p-type.
The paper is well written and presents interesting results. The following minor point should be addressed before acceptance for publication.
The ultra-low hole density epitaxial graphene Hall devices were obtained by standard high temperature sublimation of 6H-SiC(0001), followed by deposition of a protective metal layer, patterning, and removing of metal using dilute aqua regia. This final step produces a low-electron-density epitaxial graphene device (as demonstrated by the authors in a previous publication). The ultralow-hole-density graphene was obtained leaving the low-electron-density device in ambient condition over a long period of time, due to the p-type doping introduces by air and ambient humidity. The authors should comment on the stability of the value of this p-type doping level over long time aging. The monolayer epitaxial graphene grows on the stepped surface of SiC, as shown in the AFM height and phase measurements in Fig.1. Preferential formation of bilayer graphene is typically found at step edges. As a matter of fact, the monolayer-bilayer junction strongly affects the electronic transport perpendicular to its interface (see, as a reference, Phys. Rev. B 86, 235422 (2012)), and an anisotropy can be observed between devices fabricated in the direction perpendicular or parallel to the steps (see, as references, Phys.Rev. B 85, 235402 (2012) and ACS Nano 7, 3045–3052 (2013)). Did the author observe such anisotropic transport in their devices?
Author Response
Dear Editors,
Thank you for sending us the Referees’ reports on our manuscript Materials-570487 by Chiashain Chuang et al.. We would like to thank the Referees for their useful comments and appropriate suggestions on our paper. We note that both Referees recommended our manuscript to be published after revisions. We have addressed all the raised issues and have improved our manuscript in accordance with the Referees’ comments and suggestions. Therefore, we hope that our revised manuscript is now suitable for publication in Materials.
Sincerely,
Prof. C. Chuang, Prof. C.-T. Liang and Dr. R. E. Elmquist
Response to Referee 1
Possible applications in magnetic sensing, magnetoresistance devices and interesting results
The positive magnetoresistivity persisted up to 300 K, indicating possible applications of the ultralow-hole-density monolayer graphene in magnetic sensing and magnetoresistance devices. The ultralow-p-type graphene was converted into ultralow-n-type one by gentle heating in vacuum. Almost identical magnetoresistivity ratio (MR) was found in both cases, indicating that the observed large MR is an intrinsic property of an ultralow-density monolayer graphene, regardless whether it is n-type or p-type.
Comments:
The paper is well written and presents interesting results. The following minor point should be addressed before acceptance for publication.
We thank Referee 1 for useful comments and enthusiasm.
The stability of p-type doping level over long time aging
The authors should comment on the stability of the value of this p-type doping level over long time aging.
We thank the Referee for useful comments on the stability of p-type doping level over long time aging. We particularly discuss this p-type doping level over long time aging situations in the Materials and Methods.
In original Materials and Methods Section, line 94:
“Since air and moisture usually introduce p-doping, ultralow-hole-density graphene can be obtained if the low-electron-density device is left in ambient condition over a long period of time.”
In revised Materials and Methods Section, line 94:
“Since air and moisture usually introduce p-doping, ultralow-hole-density graphene can be obtained if the low-electron-density device is left in ambient condition over a long period of time (say around 3 months). However, the carrier densities of our low-carrier-density EG devices are not stable in air. Therefore, we need to measure the p-type devices in a cryostat which only exposes to helium gas/liquid without air molecule absorption so as to keep the p-type device stable in a long period of time (over two months).”
The anisotropic transport in the EG devices
The monolayer epitaxial graphene grows on the stepped surface of SiC, as shown in the AFM height and phase measurements in Fig.1. Preferential formation of bilayer graphene is typically found at step edges. As a matter of fact, the monolayer-bilayer junction strongly affects the electronic transport perpendicular to its interface (see, as a reference, Phys. Rev. B 86, 235422 (2012)), and an anisotropy can be observed between devices fabricated in the direction perpendicular or parallel to the steps (see, as references, Phys. Rev. B 85, 235402 (2012) and ACS Nano 7, 3045–3052 (2013)). Did the author observe such anisotropic transport in their devices?
We thank the Referee for useful comments on the anisotropic transport issue in the EG device. Yes, our results showed related characteristics about anisotropic transport in our devices. We particularly discuss anisotropic transport situations in the Materials and Methods.
In original Materials and Methods Section, line 79:
“We can see narrow terraces formed on the SiC substrate.”
In revised Materials and Methods Section, line 79:
“We can see narrow terraces and bilayer graphene near the step edges formed on the SiC substrate. Such terrace growing direction is almost perpendicular to the source and drain current direction since our AFM tip scanning direction is parallel to source and drain current direction, which is a better contact design due to the anisotropic quantum Hall effect in order to observe flat Hall plateau and positive magnetoresistance [25]. Interestingly, such monolayer-bilayer junctions or carbon buffer layer in EG typically reveal pronounced p-type or n-type doping ranging from 1012 to 1013 cm-2, which is highly consistent with our experimental results [26-28]. In this work, we did not get to measure the anisotropic transport perpendicular/parallel to the terrace in our EG devices which is an interesting subject warranting further investigation in the future.”
In accordance with the Referee’s comments, we have added the following references:
[25] Schumann, T.; Friedland, K.-J.; Oliveria, Jr. M. H.; Tahraoui, A.; Lopes, M. J.; Riechert, H. Anisotropic quantum Hall effect in epitaxial graphene on stepped SiC surfaces, Phy. Rev. B 2012, 85, 235402.
[26] Giannazzo, F.; Deretzis, I.; La Magna, A.; Roccaforte F.; Yakimova R. Electronic transport at monolayer-bilayer junctions in epitaxial graphene on SiC, Phy. Rev. B 2012, 86, 235422.
[27] Nicotra, G.; Ramasse, Q. M.; Deretzis, I.; La Magna, A.; Spinella C.; Giannazzo F. Delaminated Graphene at Silicon Carbide Facets: Atomic Scale Imaging and Spectroscopy, ACS Nano 2013, 7, 3045.
[28] Ristein, J.; Mammadov S.; Seyller Th. Origin of Doping in Quasi-Free-Standing Graphene on Silicon Carbide, Phys. Rev. Lett. 2012, 108, 246104.
Response to Referee 2
Possible application for magnetic sensor is an interesting finding
This manuscript reports magneto resistivity of low carrier density epitaxial graphene on SiC in a broad range of temperature and magnetic field. Carrier density below 1010 cm-2 is not easily achievable in epitaxial graphene due to the effect of the substrate. The authors presented extensive data set on this lowly p doped sample which is not well studied.
Comments:
The 100 % MR at 9 T for low doped epitaxial graphene and its possible application for magnetic sensor is an interesting finding.
We thank Referee 2 for useful comments and enthusiasm.
Comments and suggestions as below. The size of the Hall bar is important information that should be stated in method.
We thank the Referee for useful suggestions. We described the size of Hall bar in Materials and Methods section.
In original Materials and Methods Section, line 91:
“We performed standard optical lithography to fabricate Hall bar devices.”
In revised Materials and Methods Section, line 91:
“We performed standard optical lithography to fabricate Hall bar devices with source and drain length L = 600 μm and width W = 100 μm [17].”
Related to the question 1), in all figures, Rxx is presented in the unit of Ohm. Resistivity values in the unit of Ohm per square should be used to consider transport properties. In page 8, line 160, resistivity can not be compared to resistance quantum if it is not per square. Also, though I agree that the drop of Rxx to 4 kohm (fig 2) is not due to WL, Rxx is higher than h/2e can not rule out WL.
We thank the Referee for specific suggestions. We replaced all the unit of Ohm by Ohm per square in figure 4 (a), (b) and 6 in Experimental Results section. Furthermore, we replaced all the Rxx and Rxy by ρxx and ρxy so as to reveal the transport properties clearly. We also agreed the Referee’s opinions. Therefore, we revised our sentences reasonably about WL.
In original Experimental Results Section, line 183:
“However, in our case is higher than the quantum resistivity h/(2e2) so that the conventional WL model [31] is not valid.”
In revised Experimental Results, line 183:
“However, in our case dropped rapidly below 4 kΩ/â–¡ so that the conventional WL model [37] is not valid.”
In the experimental results page 3, line 108, it is written that we will get back to this effect later for Fig. 2. But, I could not find its explanation later in the manuscript. The reason why the Rxx does not reach zero and why the Rxy decrease from the quantized resistance is expected to be addressed in the manuscript. In the lowly p-doped regime, can it be related to the effect of electron-hole puddle and contact resistance?
We thank the Referee for useful suggestions. We take out “We will get back this effect later.” this sentence. We tried to explain this effect by electron-hole puddle effect.
We can observe very nice quantum Hall plateau in n-type, which is the same low contact resistance as p-type. It is possible that the contacts introduce n-doping underneath so that the p-doped device does not show good quantum Hall steps (pn junction effect).
In original Experimental Results Section, line 116:
“We will get back this effect later.”
In revised Experimental Results, line 116:
“Such peculiar behaviors could be possibly attributed to the electron-hole puddles due to very low carrier density near Dirac’s point because of the intrinsic bilayer inclusions near EG terraces so as to cause strong carrier scattering and not follow dissipationless transport behaviors [29, 30]. It is possible that the contacts introduce n-doping underneath so that the p-doped device does not show good quantum Hall steps (pn junction effect).”
In accordance with the Referee’s comments, we have added the following references:
[29] Chua, C.; Connolly, M.; Lartsev, A.; Yager, T.; Lara-Avila, S.; Kubatkin, S.; Kopylov, S.; Fal’ko, V.; Yakimova, R.; Pearce, R.; Janssen, T. J. B. M.; Tzalenchuk, A.; Smith, C. G. Quantum Hall effect and Quantum Point Contact in Bilayer-patched Epitaxial Graphene, Nano Lett. 2014, 14, 3369.
[30] Connolly, R. M.; Chiou, K. L.; Smith, C. G.; Anderson, D.; Jones, G. A. C.; Lombardo A.; Fasoli A.; Ferrari, A. C. Scanning gate microscopy of current-annealed single layer graphene, Appl. Phys. Lett. 2010, 96, 113501
4) The authors considered activated, Mott and Efros Shklovskii VRH and quantum localization for the temperature dependence of Rxx and concluded that the most likely mechanism was quantum localization as it showed lnT dependence. I agree that the plots of activation and ES VRH are not straight and the range they fits are small. But from the figures, 2D VRH (Fig.3c) looks equally good or even looks better than the lnT (Fig 3e). Then, the reasoning of lnT dependence over the 2D VRH looks weak.
We thank the Referee for useful suggestions. We changed our sentences so as to both satisfy our results and referee’s opinions. Our results followed activation behaviors in Mott and ES VRH in the high temperature regime due to thermal energy and lnT dependent localization behaviors at low temperatures.
In original Experimental Results Section, line 130:
“In all cases, one can only obtain a linear fit over a small range of temperature, suggesting that activated behavior [33], Mott variable range hopping (VRH) [34] and Efros-Shklovskii VRH [35] may not be applicable to our experimental results. The most likely transport mechanism is localization [36] as shows a lnT dependence in the low-temperature region as presented in Fig. 3 (e).”
In revised Experimental Results, line 130:
“As shown in Fig. 3 (a), activated behavior [33] is not applicable. The linear fits over a small range of temperature suggest that, Mott variable range hopping (VRH) [34] and Efros-Shklovskii VRH [35] may be applicable to our experimental results. Some evidence forthe localization transport mechanism [36] as shows a lnT dependence in the low-temperature regime is presented in Fig. 3 (e). At present, we cannot underpin the dominant transport mechanism (VRH or localization) in our p-doped EG device.”
5) In Fig.6, Rxx approaches zero and Rxy is better quantized than Fig.2 and the only difference between Fig. 2 and Fig. 4 is that it is now lowly n-doped. Discussion of any difference between lowly p-doped and lowly n-doped transport in this sample is suggested.
We thank the Referee for her/his nice suggestions. We have similar results in lowly n-doped transport results in comparison to lowly p-doped results obtained in this sample. We particularly emphasized this in related sentences.
In original Experimental Results Section, line 180:
“As shown in Fig. 2 and in Fig. 4 (a), negative magnetoresistivity (NMR) in the sense that ρxx decreases with increasing magnetic field is observed.”
In revised Experimental Results Section, line 180:
“As shown in Fig. 2 and in Fig. 4 (a), negative magnetoresistivity (NMR) in the sense that ρxx decreases with increasing magnetic field is observed in ultra-low-hole carrier density regime.”
In original Experimental Results section, line 189
“As shown in Fig. 4 (a), at T=40 K, in the low-field regime (-0.3 T £ B £ 0.3 T), there is a crossover from NMR to positive magnetoresistivity (PMR) [38].”
In revised Experimental Results section, line 189:
“As shown in Fig. 4 (a), at T=40 K, in the low-field regime (-0.3 T £ B £ 0.3 T), there is a crossover from NMR to positive magnetoresistivity (PMR) [38], which was also observed in ultra-low-electron carrier density regime [21].”
In accordance with the Referee’s comments, we have added the following references:
[21] Chuang, C.; Yang, Y.; Elmquist E. R.; Liang, C.-T. Linear magnetoresistance in monolayer epitaxial graphene grown on SiC, Material Letters 2016, 174, 118-121.
Please also see point 3 to Referee 1 in which we believe that the formation of electron-hole puddles and n-doping due to contacts in graphene may play a role.
6) The authors ascribe the negative MR in Fig. 4a to quantum transport. But quantum transport is too broad language which can mean many things. More explanation on this is suggested.
We thank the Referee for useful suggestions. We have added more explanation in the following sentences.
In original Experimental Results Section, line 184:
“The observed NMR, which shows a strong temperature dependence, is tentatively ascribed to quantum transport [38].”
In revised Experimental Results Section, line 184:
“The observed NMR, which shows a strong temperature dependence, is tentatively ascribed to quantum transport [38]. Such quantum transport and NMR behaviors could be described by Mott VRH, Efros-Shklovskii VRH and localization that were observed in our results.”
7) Mobility of the sample is suggested to be added in the manuscript.
We thank the Referee for nice suggestions. We have added the mobility information in our manuscript.
In original Experimental Results Section, line 112:
“Figure 2 shows and Rxy as a function of magnetic field at T = 1.5 K. From the low-magnetic field Hall slope, we estimate that the hole density to be p = 6.7 109 cm-2.”
In revised Experimental Results Section, line 112:
“Figure 2 shows and ρxy as a function of magnetic field at T = 1.5 K. From the low-magnetic field Hall slope, we estimate that the hole density to be p = 6.7 109 cm-2 and the Hall mobility to be μH = 2600 cm2/Vs.”
In original Experimental Results Section, line 221:
“By heating the graphene device at 323 K for 5 minutes, in a subsequent cool-down to 1.5 K, a crossover from ultralow p-type to ultralow n-type (n = 8.7 109 cm-2) occurs as shown by the Rxy data in figure 6.”
In revised Experimental Results Section, line 221:
“By heating the graphene device at 323 K for 5 minutes, in a subsequent cool-down to 1.5 K, a crossover from ultralow p-type to ultralow n-type (n = 8.7 109 cm-2 and μH = 10400 cm2/Vs) occurs as shown by the ρxy data in figure 6.”
8) In my opinion, the title is broad. Tiles such as "Magneto resistance of ultra-low-hole density epitaxial mono layer graphene grown on SiC" are suggested”.
We thank the Referee for considerable suggestions. We have changed our title to "Magnetoresistance of ultra-low-hole density epitaxial monolayer graphene grown on SiC".

Reviewer 2 Report
This manuscript reports magneto resistivity of low carrier density epitaxial graphene on SiC in a broad range of temperature and magnetic field. Carrier density below 1010 cm-2 is not easily achievable in epitaxial graphene due to the effect of the substrate. The authors presented extensive data set on this lowly p doped sample which is not well studied. The 100 % MR at 9 T for low doped epitaxial graphene and its possible application for magnetic sensor is an interesting finding.
I have comments and suggestions as below.
1) The size of the Hall bar is important information that should be stated in method.
2) Related to the question 1), in all figures, Rxx is presented in the unit of Ohm. Resistivity values in the unit of Ohm per square should be used to consider transport properties. In page 8, line 160, resistivity can not be compared to resistance quantum if it is not per square. Also, though I agree that the drop of Rxx to 4 kohm (fig 2) is not due to WL, Rxx is higher than h/2e can not rule out WL.
3) In the experimental results page 3, line 108, it is written that we will get back to this effect later for Fig. 2. But, I could not find its explanation later in the manuscript. The reason why the Rxx does not reach zero and why the Rxy decrease from the quantized resistance is expected to be addressed in the manuscript. In the lowly p-doped regime, can it be related to the effect of electron-hole puddle and contact resistance?
4) The authors considered activated, Mott and Efros Shklovskii VRH and quantum localization for the temperature dependence of Rxx and concluded that the most likely mechanism was quantum localization as it showed lnT dependence. I agree that the plots of activation and ES VRH are not straight and the range they fits are small. But from the figures, 2D VRH (Fig.3c) looks equally good or even looks better than the lnT (Fig 3e). Then, the reasoning of lnT dependence over the 2D VRH looks weak.
5) In Fig.6, Rxx approaches zero and Rxy is better quantized than Fig.2 and the only difference between Fig. 2 and Fig. 4 is that it is now lowly n-doped. Discussion of any difference between lowly p-doped and lowly n-doped transport in this sample is suggested.
6) The authors ascribe the negative MR in Fig. 4a to quantum transport. But quantum transport is too broad language which can mean many things. More explanation on this is suggested.
7) Mobility of the sample is suggested to be added in the manuscript.
8) In my opinion, the title is broad. Tiles such as "Magneto resistance of ulta-low-hole density epitaxialmonolayergraphene grown on SiC" are suggested.
Author Response

(The authors gave the same response as above.)

Round 2
Reviewer 2 Report
Dear Authors,
I believe my comments are well addressed in the revised manuscript. This manuscript can be accepted in the present form.